# Extracellular Matrix-Based Approaches in Cardiac Regeneration: Challenges and Opportunities

**DOI:** 10.3390/ijms232415783

**Published:** 2022-12-13

**Authors:** Thi Van Anh Vu, Daniela Lorizio, Roman Vuerich, Melania Lippi, Diana S. Nascimento, Serena Zacchigna

**Affiliations:** 1Cardiovascular Biology Laboratory, International Centre for Genetic Engineering and Biotechnology (ICGEB), 34149 Trieste, Italy; 2Centro Cardiologico Monzino, 20138 Milano, Italy; 3Department of Medicine, Surgery and Health Sciences, University of Trieste, 34149 Trieste, Italy; 4ICBAS—Instituto de Ciências Biomédicas Abel Salazar da Universidade do Porto, 4050-313 Porto, Portugal; 5i3S—Instituto de Investigação e Inovação em Saúde, Universidade do Porto, 4200-135 Porto, Portugal; 6INEB—Instituto Nacional de Engenharia Biomédica, Universidade do Porto, 4200-135 Porto, Portugal

**Keywords:** extracellular matrix, cardiac regeneration, mechanobiology

## Abstract

Cardiac development is characterized by the active proliferation of different cardiac cell types, in particular cardiomyocytes and endothelial cells, that eventually build the beating heart. In mammals, these cells lose their regenerative potential early after birth, representing a major obstacle to our current capacity to restore the myocardial structure and function after an injury. Increasing evidence indicates that the cardiac extracellular matrix (ECM) actively regulates and orchestrates the proliferation, differentiation, and migration of cardiac cells within the heart, and that any change in either the composition of the ECM or its mechanical properties ultimately affect the behavior of these cells throughout one’s life. Thus, understanding the role of ECMs’ proteins and related signaling pathways on cardiac cell proliferation is essential to develop effective strategies fostering the regeneration of a damaged heart. This review provides an overview of the components of the ECM and its mechanical properties, whose function in cardiac regeneration has been elucidated, with a major focus on the strengths and weaknesses of the experimental models so far exploited to demonstrate the actual pro-regenerative capacity of the components of the ECM and to translate this knowledge into new therapies.

## 1. Introduction

The human heart has a poor regenerative capacity and myocardial damage is invariably repaired by scarring, with the progressive loss of the contractile function [1]. This makes cardiovascular diseases the first cause of morbidity and mortality worldwide, with existing therapies aiming at either preventing cardiovascular events or preserving the residual function [2]. Innovative therapies capable of promoting cardiac regeneration, i.e., the de novo formation of functional myocardium, would be real game changers in this medical field. During the first day of life (P1), neonatal mice can regenerate their heart, mostly through the activation of cardiomyocyte (CM) proliferation [3]. This response is, however, readily lost after birth, being absent at P7 [3]. This regeneration-to-repair transition is associated with the physiological maturation of CMs, which undergo mitosis but fail cytokinesis. This finally results in a binucleation and subsequent cell cycle exit, with the loss of the regenerative capacity. This regenerative window of cardiac regeneration early after birth constitutes a great opportunity to dissect mechanisms that can be reactivated in the adult as part of a regenerative therapeutic strategy. 

The extracellular matrix (ECM) is the non-cellular component of all the tissues. Besides providing structural support to cells, it conveys multiple signals that influence different aspects of cell behavior, including proliferation and differentiation. Therefore, both ECM components and its mechanical properties may likely control the cardiac cell fate during the regeneration of the heart. Some concepts supporting this hypothesis are the massive remodeling of the ECM during the regeneration-to-repair transition in rodents [3,4,5] or the high regenerative potential of the zebrafish heart, which owns an ECM that is completely different from the mammalian one, throughout adult life [6].

However, the translation of this knowledge into new heart regenerative therapies is still missing. One reason may be the fact that the models used to assess the role of the ECMs’ proteins on cardiac regeneration often oversimplify the problem and display specific limitations. Furthermore, in addition to CM proliferation, cardiac regeneration requires an adequate revascularization to provide trophic support to the growing myocardium, a process that has been largely overlooked [7]. 

In this review, we discuss the evidence supporting the role of the ECM in promoting the regeneration of the myocardium, including cardiac vessels. In addition, we describe and discuss the models and experimental approaches which have been used to assess how the components of the ECM control cardiac regeneration, and we conclude that the complementary use of cell culture-based (in vitro) and in vivo approaches is required to provide convincing evidence of the therapeutic potential of the ECMs’ components and thus to consider them as targets for cardiac regeneration. 

## 2. ECM Proteins That Promote CM Proliferation and Cardiac Regeneration

Although a key function of the ECM in regulating the different phases of the development of the heart is well documented, very few studies specifically assessed the role of the proteins of the ECM in controlling CM proliferation. However, sparse evidence supports this concept. Hyaluronan, the most abundant glycosaminoglycan in the developing mouse heart, is known to provide a hydrated environment that promotes cell proliferation [8]. In addition, the hyaluronan-binding protein Hapln1 (also known as Cartilage link protein 1/Crtl1) may also promote myocardial proliferation, as Crtl1 knockout mouse embryos show severe atrial and atrioventricular septal defects due to hypoplasia [9]. While the involvement of Hlpn1 in neonatal regeneration is up to now undisclosed, a recent work showed that hapln1 defines an epicardial cellular subset necessary for CM expansion during the regeneration of a zebrafish heart. It is of interest that the genetic ablation of hapln1-expressing cells or the genetic inactivation of hapln1b results in an aberrant hyaluronic acid deposition, decreased CM proliferation, and impaired regeneration [10].

Evidence that the loss of the regenerative potential of the mouse heart soon after birth is accompanied by profound ECM remodeling has recently supported its role in this regeneration-to-repair transition. In fact, the ECM from P1 regenerative hearts is more effective in inducing CM proliferation than the ECM isolated from P7 hearts. This has been attributed to agrin, a heparan sulfate proteoglycan particularly abundant at P1, which promotes CM proliferation in vitro and is able to reactivate the CM proliferation in adult mice and pigs after myocardial infarction (MI) [11,12]. This mitogenic effect was reproduced in both mouse- and human-induced pluripotent stem cells (iPSCs), and it has been attributed to agrin binding to α-Dystroglycan (Dag1), leading to the disassembly of the dystrophin–glycoprotein complex (DGC) and subsequent myofibril disassembly and the activation of downstream signaling molecules, including Yap and ERK [11]. This evidence constitutes a proof-of-concept that ECM remodeling after birth may limit the regenerative potential of the heart and that the ECM proteins more abundant in regenerative stages are promising targets to reactivate the mechanisms of a cardiac regeneration in adulthood [5]. 

Another ECM protein claimed to promote a myocardial regeneration is periostin, which is abundantly expressed in mammals by both the developing heart and the adult myocardium after MI. In neonatal mice, periostin induces CM proliferation after MI through a PI3K/GSK3β/cyclin D1 signaling-mediated mechanism [13]. The ability of periostin to promote a CM proliferation and cardiac regeneration in adults is more controversial. On the one hand, some studies showed that periostin induces the cell cycle re-entry of adult CMs in rats and pigs after MI via the PI3K/Akt pathway [14,15], whereas other few studies came to an opposite conclusion. For instance, no difference in the number of proliferating CMs was observed in mice with either a genetic depletion or an antibody-mediated inhibition of periostin [16,17]. These discrepancies may be due to the existence of different isoforms of periostin, which might have differential effects on the CM proliferation and myofibroblast activation, as well as to different experimental conditions (i.e., the acute delivery of periostin peptide vs. chronic genetic modification). 

Parallel proteomic studies in mice have revealed a novel function for two embryonically enriched ECM proteins, Slit2 and nephronectin, in stimulating the cytokinesis of postnatal CMs both in vitro and in vivo, thus shifting from binucleation to cell cycle progression [18]. 

In addition to their direct role in promoting CM proliferation, the components of the ECM can also contribute to cardiac regeneration through alternative mechanisms. For example, in the regenerating zebrafish heart, fibronectin does not stimulate the CM division, but it allows for their mobilization and integration into the injured site [19]. 

## 3. ECM Proteins That Promote Endothelial Cell Proliferation and Cardiac Revascularization 

Angiogenesis is the main mechanism by which new blood vessels arise in adult organisms and therefore is of outmost importance for a cardiac revascularization after MI [20]. The ECM extensively contributes to this process by providing anchorage for sprouting vessels, while stabilizing the structure of the newly formed vessels [21], directly effecting the endothelial cell function and/or by serving as a reservoir of cytokines and growth factors that regulate angiogenesis. Although the role of specific ECM proteins in promoting the formation of new blood vessels has been described during development [21,22], tumor angiogenesis [23], and wound healing [24], less information is available on how the components of the ECM regulate the angiogenic processes in the heart and, in particular, on their capacity to foster a cardiac revascularization after MI. 

A series of studies indicate that the ECMs’ proteins can be exploited as novel therapies to induce angiogenesis after MI. For instance, the injection of the ECM hydrogels derived from healthy porcine myocardium improves the cardiac function and reduces adverse remodeling in both the small and large animal models of MI [25,26]. While the mechanism of this effect has remained unclear for many years, recent studies pointed to the capacity of these ECM hydrogels to increase the revascularization of the infarcted area, thereby reducing CM apoptosis and fibrosis [27]. These positive pre-clinical results led to the recent start of a first-in-man clinical trial, which demonstrated the safety and feasibility of a trans-endocardial injection of VentriGel, a ECM hydrogel derived from a decellularized porcine myocardium, in early and late post-MI patients with left ventricular dysfunction [28]. Although the study was not designed to evaluate the efficacy, there were suggestions of improvement based on the increase in the 6 min walk test distance and decrease in the New York Heart Association functional class across the entire cohort of patients. It remains unclear whether this benefit is actually due to the formation of new blood vessels and cardiac revascularization, therefore larger randomized, controlled clinical trials are needed to clarify this point.

As mentioned above, in addition to their scaffold function, the ECMs’ proteins control the formation of new blood vessels by acting as a reservoir of the growth factors and cytokines [29]. For instance, murine monocytes, which are massively recruited to the damaged cardiac area after MI, secrete endosulfatases and subsequently reduce the sulfation of heparan sulfate proteoglycans (HSPGs), a family of proteoglycans abundantly represented throughout the ECM which are able to bind to multiple growth factors. Reduced HSPG sulfation in turns releases the Vascular Endothelial Growth Factor (VEGF)-A, the main inducer of angiogenesis, through its binding to its receptor VEGFR2 on the endothelial cell’s surface [30]. 

During the heart regeneration of zebrafish, fast angiogenic sprouting, which is highly dependent on collagen guidance [31], is necessary for a CM proliferation and an effective regeneration [7]. This myocardial revascularization, which initiates in the sub-epicardium and progresses towards the endocardium, forms a vascular scaffold that is necessary for a CM replenishment during regeneration [32]. It is of note that the matricellular factor cellular communication network factor 2a (ccn2a), also named ctgfa, is required for a CM attachment to newly formed blood vessels and subsequent myocardial regeneration [33]. Another mechanism by which the components of the ECM can control the vessel regeneration after a cardiac injury in zebrafish relies on a paracrine crosstalk between coronary endothelial cells and epicardial cells, which involves the VEGF-C-induced production of the ECM protein Emilin2a, with the subsequent expression of Cxcl8a by epicardial cells and the activation of Cxcr1^+^ endothelial cells [34]. Collectively, these studies in zebrafish demonstrate that revascularization and new muscle formation are interconnected, both necessary for regeneration and equally dependent on the ECM. Similarly, angiogenesis precedes CM migration to the injury site during neonatal regeneration in mammals [35], although no study has so far specifically addressed the role of the ECM in this process.

## 4. Mechanical Properties of the ECM Controlling Regeneration

Beyond its composition, the mechanical properties of the ECM also affect cell proliferation and regeneration in the heart. The sudden loss of the CM proliferative potential after birth occurs exactly at the time when the stiffness of the ECM increases [36]. The injury response upon MI or other pathological conditions stimulates an extensive remodeling of the ECM that comprises both biochemical but also mechanical alterations [37]. This fibrotic response is frequently associated with an increase in the stiffness of the ECM that impacts the adjacent cells. The influence of the ECMs’ stiffness on the proliferative capacity of CM is as of yet unclear. While there is evidence that a higher matrix stiffness induces immature-like CM phenotypes [38], others have shown that rat and mouse CMs seeded on compliant surfaces de-differentiate and proliferate more than those seeded on more rigid structures [39]. The latter study further showed that the substrate’s rigidity primarily affects CM cytokinesis rather that karyokinesis and that chemically induced sarcomere disorganization triggers the CM cell cycle re-entry [39]. In line with this, the reduction in the stiffness of the ECM by the pharmacological inhibition of lysyl oxidase (LOX) has been shown to preserve the regenerative capacity of a mouse heart at day 3 after birth [36], suggesting that the increased stiffness of the ECM at birth may cooperate in the loss of the regenerative capacity of the mammalian heart in the early post-natal life.

At the same time, alterations in the rigidity of the ECM also influence the vessel growth. Multiple studies show that the reduced stiffness of the ECM promotes angiogenesis and cardiac regeneration [40,41,42]. Controversial results are shown by other studies, which claim that the increased stiffness of the ECM promotes a new vessel formation in the culture [43,44]. This contradiction may be explained by the different experimental settings (i.e., the use of 2D, 3D, or in vivo models), but overall, they indicate that both the composition of the ECM and mechanical properties contribute and control the growth of CMs and blood vessels during a heart regeneration.

## 5. Strategies to Study the Role of ECM Proteins in Cardiac Regeneration in Experimental Models 

As briefly summarized in Figure 1, different technologies have contributed to our current understanding of the molecular and structural ECM cues that control CM proliferation and angiogenesis and that can be exploited therapeutically. Advances in proteomics and bulk- and spatial-transcriptomics have described the changes in gene/protein expression and post-translational modifications of the ECMs’ proteins during development in zebrafish and mice [4,6,11,45] and in pathological conditions, such as acute ischemia, in both rodents and large mammals [4,46]. 

Starting from these descriptive data, a functional validation is achieved using either the whole ECM or its specific components, as described in Table 1. Bulk ECM preparations can be obtained through the decellularization of the cardiac tissue by either physical (e.g., sonication, freeze–thaw cycles) or chemical (e.g., detergents) methods that are used separately and in combination [47,48]. An optimal decellularizing protocol should result in cell removal with the preservation of a close-to-native ECM structure and composition, which can be used for either in vitro studies or an in vivo implantation. For example, a decellularized whole ECM from rat fetal hearts promoted CM proliferation to a higher extent than the ECM derived from post-natal hearts. In particular, the regenerative capacity appeared to be progressively lost with ageing, as the neonatal ECM was still more pro-regenerative than the adult one [42,49,50].

To address the role of the specific components of the ECM, the latter can be generated using recombinant DNA technology, enabling the application of individual ECM proteins to both cultured cells and animal models [11,12,14]. Alternatively, some proteins can be directly extracted from the tissues. One major example is represented by collagen I, that is commercially available as a rat tail product and it has been exploited in several studies [55,56].

The functional role of either the whole ECM or its specific components can be validated in vitro and in vivo, often entailing inter-species applications. This is possible because of the high homology among mammalian genomes (i.e., rat agrin induces the proliferation of both mouse primary CMs and human iPSC-derived CMs) and because a complete cell and DNA removal upon decellularization may avoid most problems of immunogenicity and rejection [11,41,49,57]. Yet, the expression of xenogeneic epitopes may represent a big hurdle. For example, the carbohydrate galactose-α 1,3-galactose (alpha-gal) is expressed in all mammals except for primates, causing severe immune responses when the matrix scaffolds derived from pigs are injected in humans [58]. Further studies are needed to overcome this limitation.

### 5.1. In Vitro Models

As cardiac regeneration requires the formation of both new muscle tissue and new vessels, multiple in vitro models have been generated to assess the effect of the ECMs’ proteins on both CMs and endothelial cells.

One challenging objective is the induction of cell cycle progression in primary CMs. Considering the absence of the proliferative potential of adult CMs in mammals, most studies adopted neonatal CMs harvested from rodents within the first week of life, when the cardiac regenerative capacity is partially preserved [36]. More recently, numerous approaches have exploited human iPSC-derived CMs, although the majority of these studies did not assess the role of the ECM on CM proliferation, but they rather focused on CM maturation and the generation of fully differentiated cardiac-like tissues [57]. In parallel, the potential of the ECM to induce neo-angiogenesis has been mainly tested in primary endothelial cells, most often human umbilical vein endothelial cells (HUVEC), and iPSC-derived endothelial progenitors [12,40,52]. Notably, one aspect to keep in mind is that endothelial cells are quite heterogeneous depending on the tissue source, and these models rarely used heart-derived ones in order to investigate cardiac regeneration [59]. In addition, the ECM can indirectly stimulate the production of pro-angiogenic factors on cardiac fibroblasts, thus further contributing to the formation of a new vessel after MI by stimulating the production of the fibroblast growth factor-2 [41].

The three cell types that mainly compose the heart (CMs, endothelial cells, and fibroblasts) can be cultured individually or combined in either 2D co-cultures or 3D tissue-like structures (Figure 1). 

Despite their apparent simplicity and naiveness, 2D approaches present numerous advantages. First, they are relatively inexpensive and highly reproducible, suitable for imaging and high throughput screening. Second, the proliferation rate of both CMs and cardiac endothelial cells is usually higher in 2D cultures compared to 3D tissues, which require a long-term differentiation. Thus, numerous studies evaluating the role of the ECMs’ proteins on cardiac cell proliferation have used 2D models, in which the components of the ECM are available to the cells in the solution as a supplement to the cell culture media, or as a coating after the adsorption to the well surface. In these studies, specific components of the ECM were tested in the form of recombinant proteins or lyophilized/cryo-pulverized dECM [11,50,57]. Furthermore, reconstituted dECM hydrogels can be either added as a chemoattractant in transwell culture systems to assess the cardiac cell proliferation and migration or used as a support in vasculogenesis assays [41,49]. 

Differently from the 2D assays, 3D systems preserve and reproduce spatial cues that may direct cell growth and migration. One example is represented by cardiac organoids [60], which have been largely used to study the process of the maturation of iPSCs into a mature cardiac structure [61,62,63]. Despite their potential, cardiac organoids have been poorly exploited so far to study the effect of the ECMs’ proteins on cardiac regeneration. 

An alternative approach consists of the use of the dECM as a 3D scaffold for a subsequent recellularization with multiple cell types, that are expected to migrate and distribute following the original spatial organization of the matrix [64,65]. This could be an advantage to promote the formation of vessel-like structures within 3D cardiac tissues, a major limitation of organoids, which are usually not vascularized. More recently, 3D bioprinters allow for the precise control of a cell and matrix distribution within the tissues generated ex vivo. Bioinks are indeed composed by specific cellular elements mixed with the components of the ECM, and thus they represent a powerful tool to precisely assess the role of the ECM on cardiac cell proliferation and cardiac tissue regeneration [51,52,66].

### 5.2. Animal Models

#### 5.2.1. Zebrafish

The relevance of zebrafish as a model of cardiac regeneration stems from its capacity to regenerate new myocardium in response to damage throughout its life. This happens through the partial de-differentiation and proliferation of existing CMs, without the involvement of any stem cell population [67]. In addition, it has a short breeding and developmental time and it is easy to handle and to genetically manipulate [68]. Finally, over 70% of human genes have their orthologs in zebrafish, making this species a useful model for cardiac regeneration studies [69]. 

Multiple approaches can be used to study cardiac regeneration in zebrafish. The genetic manipulation of zebrafish allows for the generation of organisms either overexpressing or lacking specific ECM components, such as different VEGF isoforms that are retained in the ECM and that are essential for new vessel formation during heart regeneration [34]. On the other hand, as mentioned before, the whole ECM from zebrafish can be used for inter-species experimental applications. For example, it can be injected in the form of a hydrogel in the damaged myocardium of rodents after MI. This promotes CM proliferation, vasculogenesis, and it improves the cardiac function, particularly when the ECM is collected from the fish hearts subjected to a cryoinjury [49]. More than having a therapeutic applicability, this work shows that the components of the ECM are conserved across species and that the dissection of the ECM-mediated regenerative mechanisms in zebrafish may unveil the novel molecules of translational relevance.

#### 5.2.2. Rodents

Rodents, and particularly mice, are frequently used in experimental research, as they show a >99% homology with the human genome [70], a fast reproductive cycle, and they can be easily housed and manipulated in laboratory settings. As mentioned before, heart regeneration in mice is limited to the neonatal stage [3,71]. 

The first approach to study the role of specific ECM proteins on the heart regeneration in rodents is their genetic depletion, which can be either constitutive or conditional (using the Cre-Lox technology) [11,13,16]. For example, the conditional knockout of agrin in mice blunts the CM proliferation and promotes sarcomeric organization during the first week after birth [11]. Conversely, the knockout of some extracellular enzymes responsible for the post-translational modifications of the ECMs’ proteins inhibits a vessel regeneration and cardiac repair after MI, due to the altered 3D structure of the ECM and changes in the docking sites for the soluble factors [30].

Therapeutically, the whole ECM can be either sewed/glued onto the epicardial surface or injected into the infarcted region. The most common preparation consists of liquid hydrogels, although solid ECM microparticle formulations may increase the functionality by extending the stability and controlling the release of the ECMs’ proteins [53]. To shed light on the mechanism by which the whole ECM impacts on cardiac regeneration, it is possible to administer chemical inhibitors that are expected to interfere with the relevant pathways. This was the case for ErbB2 receptor inhibitors, which were delivered systemically after the local injection of zebrafish ECM in the murine infarcted heart, and they significantly impaired the regeneration [49]. In addition, the effect of the ECMs’ structure and stiffness can be assessed by administering ECM cross-linking modulators, such as BAPN or glutaraldehyde [41,42]. Finally, individual ECM components can be directly injected into the damaged heart to assess their effect on the regeneration, as done in the case of agrin and periostin [12,14,41]. The key results of these studies are summarized in Table 1.

Besides stimulating an endogenous regeneration, the ECM has been used in combination with human cardiac progenitor cells (hCPCs) to create a patch for myocardial repair. While the initial aim of this approach was to promote the retention of hCPCs, subsequent studies ascribed the main therapeutic effect to the presence of the ECM, which, per se, promoted tissue remodeling and preserved the cardiac function in a rat model of right ventricular failure [51,58,66].

It is noteworthy that there are rodent strains that show an enhanced capacity to regenerate and that could expand the option to study the role of the ECM in cardiac regeneration. One example is the MRL mouse, a strain that manifests the autoimmune disorders mainly caused by the genetic alteration of the Fas gene [41,42]. Notably, these mice are characterized by a fast healing, likely explained by several genetic loci that are associated with this phenotype. Heart regeneration has been reported in this model, at least in certain conditions [72], but no studies specifically assessing the role of the ECM have been performed so far. A second example is the spiny mouse (*Acomys cahirinus*), which shows wound healing without scarring in different organs, including the heart upon MI [73]. These models provide helpful systems to unveil the ECMs’ role in cardiac regeneration in adult mammals. 

#### 5.2.3. Pigs

Despite the high clinical relevance of large animals, only a few studies have assessed the therapeutic activity of the ECMs’ proteins in MI pig models. For example, the administration of agrin resulted in being cardioprotective and pro-regenerative [12], whereas periostin promoted the formation of new myocardial strips with an improved cardiac function, but it increased fibrosis [15]. Whether this discrepancy may be explained by the different route of administration of the two proteins (antegrade infusion into the coronary artery in the case of agrin and controlled release in the pericardial space in the case of periostin), it remains an open question which deserves further investigation.

## 6. Conclusions and Future Perspectives

Here, we reviewed the existing evidence supporting the role of the ECMs’ proteins in promoting a cardiac regeneration. 

This concept arises from the demonstration that both the composition and the mechanical properties of the ECM change dramatically between regenerative and non-regenerative conditions, for example, between zebrafish and mammalian hearts or between pre-natal and post-natal hearts. To date, ‘omic’ technologies have generated a detailed description of the ECMs’ composition and its relative changes in different experimental settings. While proteomics provides a realistic description of the actual composition of the ECM, it often fails in detecting the changes in the proteins’ isoforms, for instance, those generated by alternative splicing as in the case of fibronectin [72]. Thus, the information derived from both the RNA and protein analysis needs to be analyzed in an integrated manner to maximize the chance to detect numerous, yet significant, differences. 

Perhaps, the most direct and striking evidence that the ECM can be exploited for regarding heart regeneration is that the implantation of the ECM harvested from either zebrafish or fetal mammalian hearts elicits a regenerative response in an injured adult myocardium. Despite being interesting for a basic investigation, this approach presents obvious limitations in terms of the translatability into a clinical scenario. On the one hand, the use of an ECM from other species might induce a potentially dangerous immune response [74]. On the other hand, the purification of the ECM from human fetuses has important ethical and technical limitations, particularly related to the availability of donors. These drawbacks could be overcome by using an autologous ECM from different organs, which has been recently considered for the generation of functional vascularized myocardial patches by 3D bioprinting [75].

In addition, some studies indicate that even the adult ECM in mammals retains some regenerative potential, which can be further enhanced by partial enzymatic digestion [54]. As mentioned above, one of the few ECM-based approaches that have reached the clinics is Ventrigel, a hydrogel derived from decellularized porcine myocardium. It is interesting to note that Ventrigel is significantly softer than the real adult cardiac ECM, which suggests that the ECMs’ stiffness, rather than its composition, may be the most important feature in exerting a pro-regenerative effect [76]. More studies which focus on the mechanical properties of the ECM are warranted to shed light on this aspect.

An additional outstanding question is whether it is better to implant the whole ECM or its individual components, which can be produced and administered as recombinant proteins. While the second approach is more straightforward and can be easily standardized, the whole ECM has the unique property to interact with and possibly stimulate all the cell types needed to build a new myocardium. Indeed, while most of the approaches have so far aimed at stimulating either CM proliferation or the formation of new blood vessels, an ECM-based biotherapeutic might have the potential to simultaneously drive both processes. 

Finally, a fundamental issue that still needs to be clarified is the most efficient mechanism that can regenerate the heart in vivo. Many studies have tried to either implant exogenous cardiac progenitors or to activate a putative population of cardiac stem cells. These approaches have been harshly questioned and, in any case, they have failed in providing a benefit in patients [2]. More recently, the idea of stimulating an endogenous regenerative response through the partial de-differentiation of existing CMs and their subsequent proliferation, like what happens in zebrafish, has provided promising results in small and large pre-clinical models [77,78]. However, the same studies also highlighted the need to keep the CM proliferation under control to avoid their overgrowth and the formation of tumor-like structures [77,78]. Although the role of the ECM has not been investigated in this case, the theory of the ‘tissue organization field’, by which the neoplastic phenotype of cancer cells is dictated by their interaction with an abnormal stroma, suggests that the ECM could provide the signals that stop the CM proliferation and promote their terminal differentiation once the regenerative process is completed [79,80]. 

In conclusion, multiple approaches have been used to claim the pro-regenerative role of ECM components. While in vitro experiments provide clearcut and straightforward readouts to elucidate the effect of the ECM proteins on cardiac cells (i.e., discrimination between CM proliferation and cardioprotection), in vivo studies appear essential to prove a therapeutic benefit. As highlighted in Table 1, the regenerative properties of very few ECM proteins have been confirmed by both in vitro and in vivo models. In most instances, only one model has been implemented or multiple models have led to conflicting results. For example, periostin was able to promote CM proliferation in some, but not all, in vitro studies, possibly depending on its isoforms, and unwanted effects, such as fibrosis, emerged during in vivo studies. Thus, we conclude that multiple experimental models, both in vitro and in vivo, need to be considered and used to provide convincing evidence of the pro-regenerative capacity of the ECM, before considering it an effective strategy for cardiac regeneration.

## Figures and Tables

**Figure 1 ijms-23-15783-f001:**
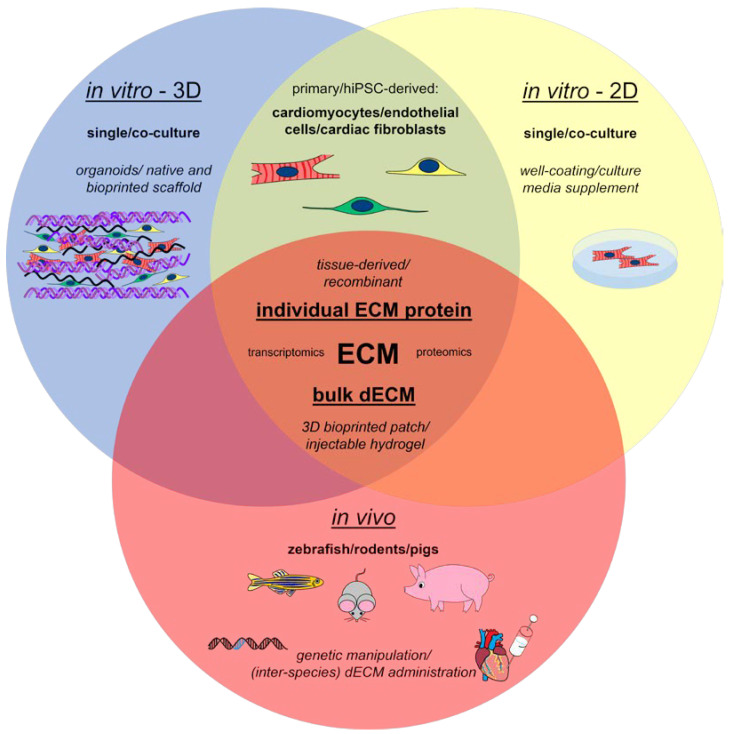
Overview of the main approaches exploited to investigate the role of ECM in cardiac regeneration.

**Table 1 ijms-23-15783-t001:** ECM proteins described to have a pro-regenerative effect in the heart.

ECM Protein	In Vitro/ In Vivo	2D/3D	Cell Type	Species	Method	Effect of the ECM Protein and Mechanism	Ref.
Hyaluronic acid	in vivo			zebrafish	hapln1 KO	↑ CM proliferation	[10]
	in vivo			mouse	hapln1 KO	↑ CM proliferation	[9]
Periostin	in vitro + in vivo	2D	neonatal CMs	mouse	periostin KO	↑CM proliferation by activation of PI3K/GSK3β/cyclin D1 pathway, ↑ angiogenesis	[13]
	in vitro + in vivo	2D	adult CMs	rat	recombinant periostin	↑ CM proliferation by activation of integrins and PI3K pathway, ↑ angiogenesis	[14]
	in vivo			pig	recombinant periostin	↑ angiogenesis and fibrosis	[15]
	in vitro + in vivo	2D	neonatal CMs and FBs	rat, mouse	periostin KO, overexpression and recombinant periostin	No changes in cell proliferation	[16]
	in vitro + in vivo	2D	cardiac FBs	rat	anti-periostin antibodies	↑ fibrosis by TGF-β1 activation	[17]
Agrin	in vitro + in vivo	2D,3D	hiPSC-CMs, neonatal mouse CMs	mouse, human	recombinant agrin and agrin cKO mice	↑ CM proliferation by DGC disassembly and Yap/ERK pathway	[11]
	in vitro + in vivo	2D	HUVECs	pig, mouse, human	recombinant agrin	↑ cardiac function and angiogenesis	[12]
Slit2, Nephronectin	in vitro + in vivo	2D	neonatal CMs	rat	recombinant proteins	↑ CM cytokinesis	[18]
Fibronectin	in vivo			zebrafish	fibronectin KO	↑ CM mobilization	[19]
HSPGs	in vivo			mouse	sulfatase-1 and -2 KO and overexpression	↓ angiogenesis by reduced VEGF availability	[30]
Ccn2a	in vivo			zebrafish	ccn2a KO, overexpression	↑ CM proliferation by activation of Tgfβ/pSmad3 pathway	[33]
Emilin2a	in vivo			zebrafish	emilin2a KO, overexpression	↑ CM and EC proliferation by activation of cxcl8a-cxcr1 pathway	[34]
Whole ECM	in vivo			pig	porcine dECM hydrogel	↑ cardiac function	[25]
	in vivo			rat	porcine dECM hydrogel	↑CM proliferation	[26]
	in vivo			rat	porcine dECM hydrogel	↑ CM viability and angiogenesis	[27]
	in vivo			human	Ventrigel (porcine)	Undefined (safety proven)	[28]
	in vitro + in vivo	2D	cardiac FBs, HUVECs	rat	CorMatrix (porcine)	↑ angiogenesis	[41]
	in vitro + in vivo	2D	neonatal CMs	rat	rat dECM	↑CM proliferation	[50]
	in vitro	2D	cardiac progenitors	human	zebrafish dECM	↑ CM proliferation and angiogenesis byErbB2 pathway	[49]
	in vivo			rat	porcine dECM hydrogel	↑ angiogenesis	[51]
	in vitro + in vivo	2D	HUVECs	human, rat	porcine dECM hydrogel	↑ angiogenesis	[52]
	in vivo			mouse	dECM microparticles	↑ CM proliferation	[53]
	in vitro	2D	neonatal CMs	rat	enzymatically digested rat ECM	↑ CM proliferation	[54]

Abbreviations (not explained in the main text): FBs: fibroblasts, EC: endothelial cells, dECM: decellularized ECM, cKO: conditional knock-out.

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
