# Peer review of "Extracellular Matrix-Based Approaches in Cardiac Regeneration: Challenges and Opportunities"

_ijms, 2022, doi:10.3390/ijms232415783_

Round 1
Reviewer 1 Report
The manuscript is dedicated to describe the role of ECM and its components on cardiac development or regeneration. The review is written about everything often by using generous not informative sentences that hided a real important information for the readers.
1. The meaning of the brackets in the title of the manuscript is not clear.
2. The main goal and structure of the manuscript is missing. The manuscript in many parts is written as a novel and very often sentences are not informative. The manuscript is too brad and therefore incomplete written.
3. Studies in vitro and in vivo should be separated as well as human, animal and other model systems studies. Now in many places everything is mixed into one and not clear what the authors are talking about. The ECM components are not described properly. The section titles also should clearly represent the goal of the section. For example: 2.2 ECM proteins that promote CM proliferation and cardiac regeneration. The question is – human, animal or other types of the model systems in vivo or in vitro? The same inaccuracies are all over the manuscript.
4. The type and origin of the used cells or model systems in vivo or in vitro should be clearly stated. For example, Page 3. Line 100-1001. The sentence “Hyaluronan, the most abundant glycosaminoglycan in the developing heart, is known to provide a hydrated environment that promotes cell proliferation”. It is not clear neither the heart origin, nor the type of the cells. Such not informative sentences are all over the manuscript.
Another example. Line 200-201 - “For example, the whole ECM from fetal, neonatal and adult hearts differentially affects cardiac cell proliferation and regeneration, both ex vivo and in vivo [9, 40b, 45].” In what origin of the hearts? What origin and type of the cells? What means “differently” affected? Such sentences are about everything and nothing and are not informative.
5. Figure 1. The scheme looks strange, i.e. the iPS and other cells are on the top but do not belong to the 2D and 3D systems in vitro? In that case where are they tested?
6. The Tables with summarized and informative data are missing.
7. The specific conclusions are missing.
Author Response
Dear Referee,
Thank you for your constructive and helpful comments, which we have used to improve our manuscript. Here is a point-by-point response to your comments.
- We agree on the possible confusion generated by our original title. We have now changed it, to make it more informative about manuscript content.
- We apologize if a clear structure seems missing. We are aware that several reviews have extensively described the composition of cardiac ECM and its changes during development and upon injury. Thus, we have now removed the first part on developmental changes, and we added a chapter about ECM mechanical properties that have been demonstrated to affect cardiac regeneration, beyond the previous chapters describing some examples of ECM components that alter CM and endothelial cell proliferation, respectively. As you pointed out, every group has used its own method and experimental set-up. Exactly for this reason, our idea was to focus on the multiple experimental approaches used so far to investigate and demonstrate the therapeutic potential of cardiac ECM, and of its components, in driving cardiac regeneration, to highlight the strengths and weaknesses of each approach and to conclude that both cell culture and in vivo data are needed to claim the capacity of ECM components to induce cardiac regeneration. We have now extensively revised our manuscript to make this vision clearer.
- In the first part of the review, we provided a broad overview on the different proteins described to have a role in cardiac and vascular regeneration, including all species and independent of the experimental approach used. Taking the Reviewer’s suggestion, we have now added a few details on the species and model for each approach. The second part, instead, was already structured to describe the different models, by discriminating between in vitro and in vivo models, and, within animal models, every species. This is also better specified in Table 1.
- As reported in our previous answer, we have added these missing pieces of information throughout the text, when we felt they provided essential informative details.
- We apologize for the lack of clarity. We have modified the figure to clearly show that primary and iPS cells are used for both 2D and 3D models.
- Following Reviewer’s indication, we have added a Table that summarizes the existing evidence on the pro-regenerative effect of each ECM protein (or the whole matrix), with evidence of the species in which it has been tested and the model(s) used by the different authors. We used this table also to draw our final conclusions, namely that both in vitro and in vivo evidence has to be obtained to claim for a pro-regenerative effect of ECM components.
We wish to thank this Reviewer for his/her constructive comments. We are confident that the extensive modification and integrations added to our manuscript have significantly improved its quality, and that it is now acceptable for publication.
Reviewer 2 Report
The manuscript described and reviewed various aspects on the role of ECM proteins and their role in cardiac regeneration. The manuscript is well written and comprehensive. However, there are few concerns
1. Please cite lines 33-41
2. Please cite 52-57
3. Including a figure showing the role of ECM proteins in cardiac development and regeneration will be useful
4. Please include a Table comparing and describing the role of various ECM proteins in cardiac regeneration between different species as included 1) Zebrafish, 2) mice model, and 3) large animal. Please also include a table including comparative summary of the signaling pathways/proteins/other factors involved in development, differentiation, proliferation, transdifferentiation, and regeneration between species along with the evolutionary aspects.
5. Some of the recent studies are missing like https://elifesciences.org/articles/69028, https://www.sciencedirect.com/science/article/pii/S153594762031776X, https://onlinelibrary.wiley.com/doi/10.1002/adhm.202102265,
6. A similar article has discussed the role of ECM in cardiac regeneration, please see PMID: 33511134
Author Response
- We have inserted missing citations.
- We have inserted missing citations.
- In response to Reviewer 1 and because several reviews have already addressed the role of ECM proteins in cardiac development and regeneration, we have removed the first part (on development) and expanded the second part, which is more focused on the in vitro and in vivo models, so far considered and used to demonstrate the role of ECM proteins in cardiac regeneration. Thus, we feel that the suggested figure would result out of topic in this new version.
- Following Reviewer’s suggestion, we have included a Table that summarizes the existing evidence on the pro-regenerative effect of each ECM protein (or the whole matrix), with evidence of the species in which it has been tested and the model(s) used by the different authors. We used this table also to draw our final conclusions, namely that both in vitro and in vivo evidence has to be obtained to claim for a pro-regenerative effect of ECM components.
- We agree that the suggested references are informative and we included them in our bibliography.
- We are aware of previous work discussing the role of ECM in cardiac regeneration and the reference indicated in your comment was indicated in our bibliography (3c). Exactly for this reason, we have tried to provide a different perspective by focusing more on methodological aspects.
We wish to thank this Reviewer for his/her constructive comments. We are confident that the extensive modification and integrations added to our manuscript have significantly improved its quality of the manuscript, and that is now acceptable for publication.
Round 2
Reviewer 1 Report
The manuscript has been significantly improved and corrected according to the remarks. The manuscript can be published.
Reviewer 2 Report
Thank you for modifying the manuscript. Changing the color scheme of the first figure will help to read the text. Please consider.